# Measuring individual worker output in a complementary team setting: Does regularized adjusted plus minus isolate individual NBA player contributions?

**Shankar Ghimire**[1]*, **Justin A. Ehrlich**[2], **Shane D. Sanders**[2]

**1** Department of Economics and Decision Sciences, Western Illinois University, Macomb, IL, United States of America, **2** Department of Sport Management, Falk College, Syracuse University, Syracuse, NY, United States of America

* SP-Ghimire@wiu.edu

## Abstract

Adjusted plus minus (APM) measures have redefined our understanding of player value in basketball and hockey, where both are team games featuring player productivity spillovers. APM measures use seasonal play-by-play data to estimate individual player contributions. If a team's overall score margin success is figuratively represented by a pie, APM measures are well-designed to slice the pie and attribute individual contributions accordingly. However, they do not account for the possibility that better players can increase the overall size of the pie and thus increase the size of the slice (overall APM value) for teammates. Herein, we use data from NBA player-season Real Plus Minus (RPM)—a leading APM measure— for all recorded player-seasons from 2013–19 and player lineup data to test whether RPM is related to teammate quality. We run sets of linear fixed effect regression models to explain variation in RPM across player-seasons. We also employ a two-stage least square (2-SLS) method for robustness check. Both empirical approaches address potential endogeneity in the relationship of interest. We find strong evidence that RPM is related to on-court teammate quality. Despite adjusting for teammate and opponent quality, RPM does not control for complementarity effects. As such, RPM is not suited for out-of-sample prediction.

## I. Introduction

Adjusted plus minus (APM) measures have redefined our understanding of player value in basketball and hockey, where both are team games featuring player productivity spillovers [1]. In basketball, a player's APM value represents his marginal effect on the score margin per 100 possessions as compared to a league average player. It has become a leading measure for comprehensive player analysis [2–8]. The measure utilizes ridge regression in an effort to isolate individual contributions to average team score margin differentials per 100 possessions [9]. We extend the analysis by using ESPN's estimated values as explanatory variables in a set of

**Data Availability Statement:** Data are publicly available from: https://stats.nba.com/

**Funding:** The author(s) received no specific funding for this work.

**Competing interests:** The authors have declared that no competing interests exist.

fixed effects and the two-stage least square (2-SLS) regressions that seek to explain player-season APM variation.

In an ESPN article introducing the real (adjusted) plus minus measure, Steve Ilardi [8] highlights the serious flaw in the much familiar unadjusted +/- statistics: each player's rating is heavily influenced by the play of his on-court teammates. APM measures were created to disentangle setting-of-play spillovers and render player value measures that are adjusted for teammate and opponent quality and thus (purportedly) unrelated to teammate quality.

In a related paper, Brian McDonald points out that APM values seek to disentangle player marginal effects from one another by using lineup variation across a season of play and are thus commonly interpreted as teammate independent [10]. It is the case that regularized adjusted plus minus measures feature (asymptotically) *consistent* estimators of player value toward an understanding of marginal player contributions from the season that occurred. From this, however, can we interpret the measure as providing teammate-independent player contribution values? Let us think of the value that a player creates—along with his lineup teammates—as a pie whose size represents (score margin) performance. APM measures can divide each such pie into constituent slices that represent consistently estimated player contributions. Due to the asymptotic property of consistency, that is, APM is capable of dividing credit at the player level based on the lineups and performances in which the player actually played. Is this division reliable when considering the player's (counterfactual) productivity "out of sample"? Consider a teammate all of whose lineup teammates improve in ability (i.e., in every lineup in which the player plays). In this case, we know that the overall pie with which the player is associated has become larger, while the individual player retains the same player attributes. Do we expect this player to maintain the same expected APM value? More generally, do we expect player APM values in general not to be influenced by out of sample changes in playing conditions? If we are to treat the APM measure as teammate independent, then such an outcome must obtain. We present this analysis herein.

The remainder of the paper is organized as follows: Section II provides a summary of the data and their visualizations; Section III describes the estimation approach; Section IV presents the empirical results; Section V discusses the implications of the results; and Section VI concludes.

## II. Data summary and visualizations

For out of sample shifts in teammate quality that cause changes in the size of a figurative lineup performance pie, do we expect the size of a player's slice to remain the same? Rather than simply a theoretical consideration, this question has substantial empirical bearing in the modern NBA, which features a high level of player entry and player movement from year to year. Often, players change teams and are presented with near wholesale upgrades or downgrades in terms of lineup-teammate quality. To test whether APM is teammate-independent, we take advantage of frequent player movements in the NBA to determine whether a representative player's season-level APM values are related to the (minutes-weighted) average APM value of his lineup teammates in the same season. We do so using a six-year panel data of NBA player-season level ESPN *real plus minus* data and corresponding lineup data from NBA.com. The two databases are available at https://www.espn.com/ and https://stats.nba.com/ respectively. We also control for time variant features of the panel data (e.g., age and age squared). To the authors' knowledge, ESPN's *real plus minus* represents the only publicly available dataset of NBA player regularized APM values, where regularized adjusted plus minus estimation has become a standard methodology for NBA player analysis [2, 5–10]. In the remainder of the paper, we use the terms *real plus minus* and *APM* interchangeably, as real plus minus is, in fact, an APM measure.

The present paper considers whether APM measures (e.g., ESPN real plus minus) are team-mate dependent. It does so by considering player movement within and entry into the NBA. The following plots summarize the level of mobility within the NBA over the data period. Specifically, Fig 1 considers the proportion of NBA players who played for at least two teams during the 2-season period that ended with the season listed on the x-axis.

Fig 1 demonstrates that there is a high level of mobility in the NBA. Typically, more than a third of League players change teams at least once over a given two-year window. Given this level of mobility, changes in teammate ability from season to season are pervasive in the NBA. Many players explicitly change teams, and the players who do not change teams typically receive several new teammates from one season to the next. The following directed graph visualizes player flows in the NBA over the data period.

Within the directed graph (Fig 2), the direction of player flow is depicted by color-coding. The color of an edge is coordinated with the predecessor node (original team), and the edge flows to the successor node (latter team). The thickness of an edge indicates the number of players flowing along an edge over a data period. The graph demonstrates that NBA teams are very connected through the player market and that player turnover for teams is therefore high. An additional source of lineup change in the NBA is player entry into the NBA. Fig 3 demonstrates the proportion of rookie players in a given NBA season.

NBA teams have relied heavily on rookie players in recent years. In each of the last three NBA seasons, more than one-fifth of the league players have been in their rookie season. Hence, player entry into the league has also been a substantial source of lineup disruption from season to season over the sample period.

While there is substantial player movement in the NBA, this matters only if players exhibit significant heterogeneity in terms of productivity (i.e., real plus minus value). The following plots represent density plots of player-season real plus minus values for the sample. Fig 4

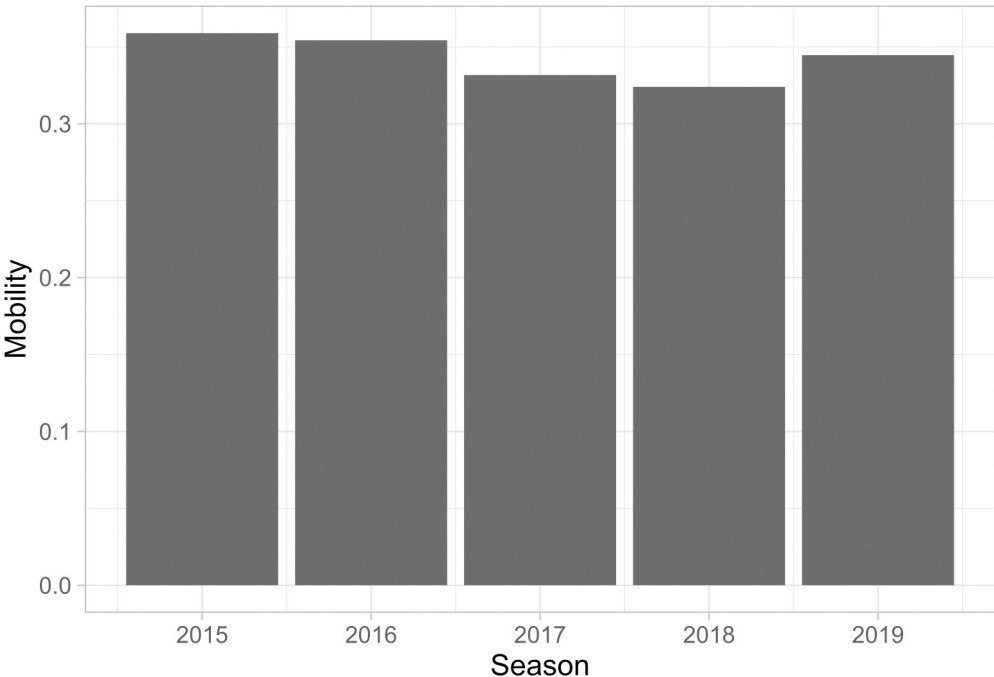

**Fig 1. Mobility in the NBA.**

**Player Flow Through Leagues**

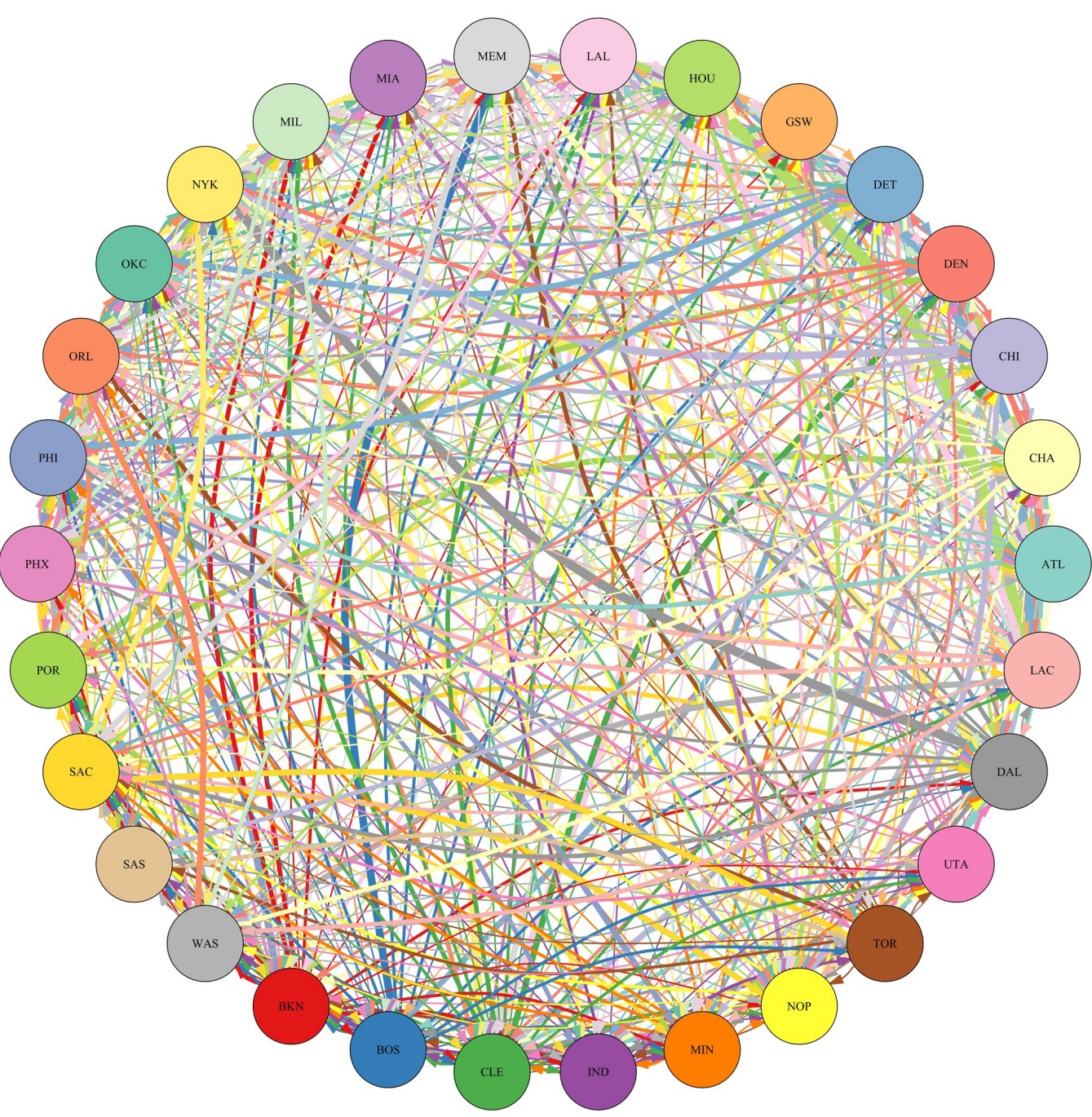

**Fig 2. Player flows within the NBA.**

represents an overall player-season density plot, while Fig 5 represents a set of color-coded density plots by position-of-play.

As these plots demonstrate, there is substantial player ability heterogeneity at each position in the NBA. While each plot is roughly bell-shaped, we observe substantial dispersion in player

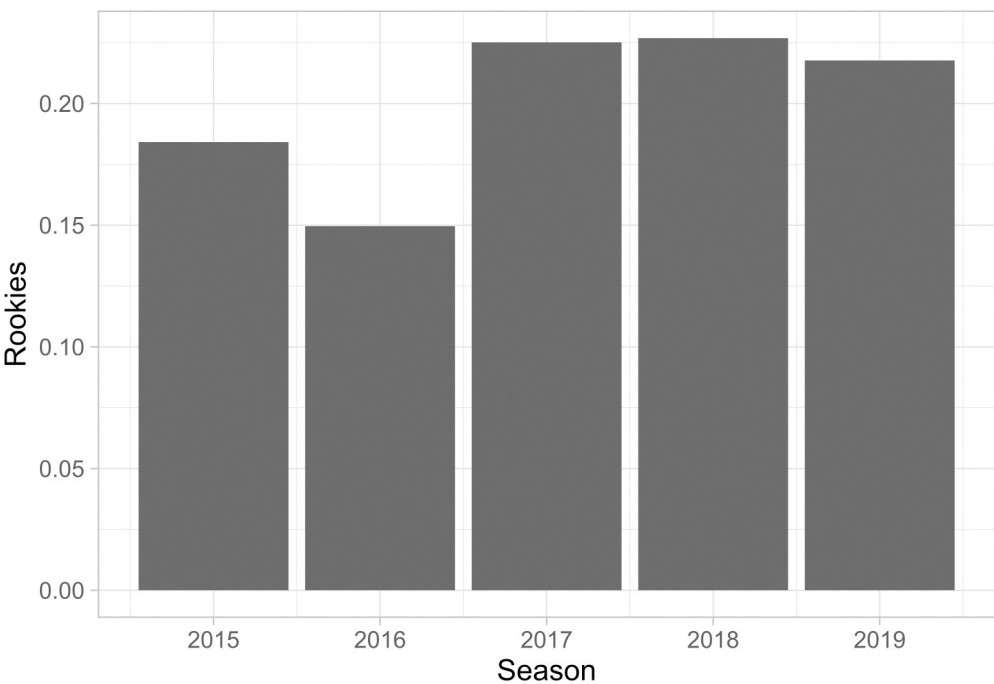

**Fig 3. Proportion of players who are rookies.**

productivity levels. In terms of range, the most productive players have an estimated +10 point score margin effect per 100 possessions, while the least productive have an estimated -7 point score margin effect. Moreover, substantial heterogeneity is observed at each position. The

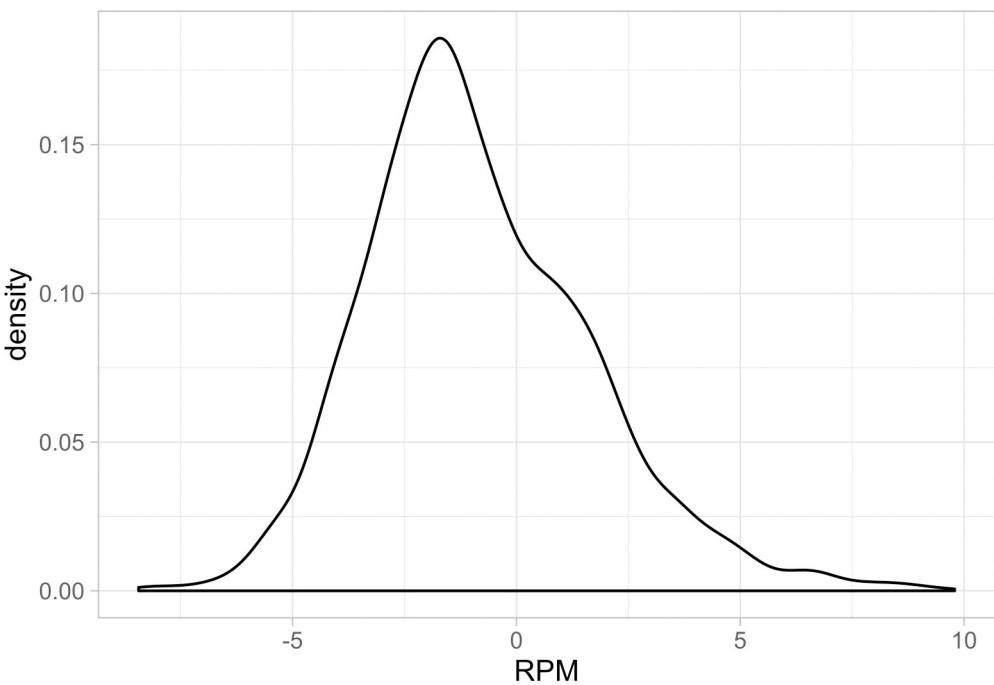

**Fig 4. Real plus minus density plot for all NBA players, 2013–14 through 2018–19.**

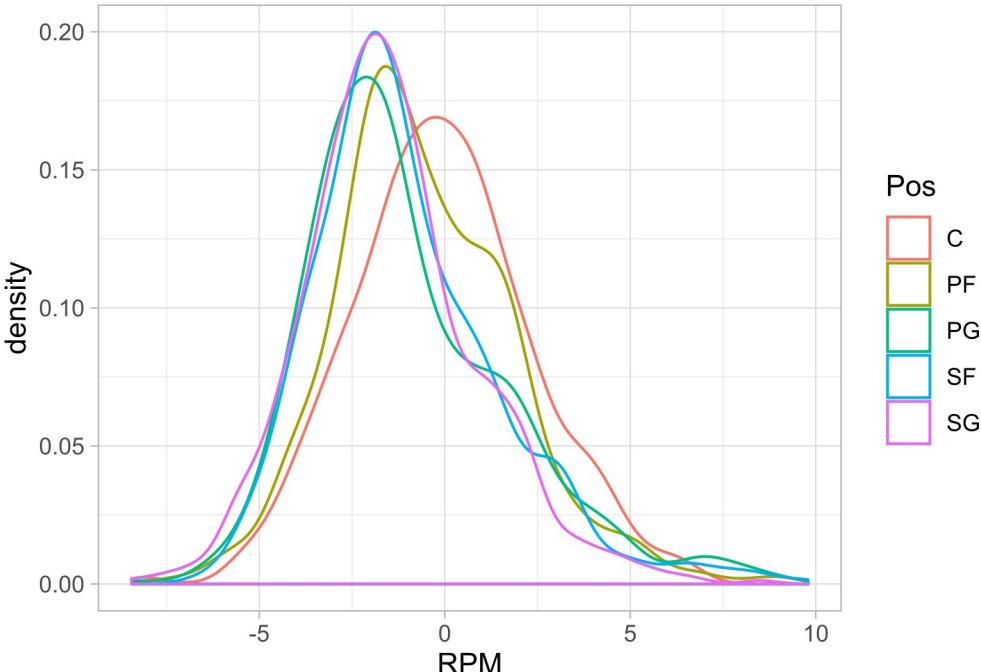

**Fig 5. Real plus minus density plots by playing position, 2013–14 through 2018–19.** The RPM densities are plotted for the Point Guard (PG), Shooting Guard (SG), Power Forward (PF), Small Forward (SF), and the Center (C) positions.

following real plus minus summary (Table 1) further demonstrates the level of player productivity (RPM) heterogeneity and of average lineup-teammate productivity (OtherPlayersRPM) heterogeneity in the NBA.

Note that the average player RPM is substantially negative because player-season observations are not weighted by minutes played within the sample. However, OtherPlayersRPM is weighted by teammate minutes played and is therefore closer to zero. Such a weighting would impose a zero-average constraint upon the variable. Even if a player is able to maintain a stable set of lineup-teammates across seasons, aging effects can change teammate ability levels fairly quickly in the NBA. Fig 6 plots the estimated aging curve (i.e. productivity profile) for NBA players over the sample period. On the y-axis is real plus minus and on the x-axis is age.

Fig 6 demonstrates that aging affects estimated player productivity substantially in the NBA. As such, the productivity of lineup-teammates can change from year-to-year even without substantial roster turnover. In our fixed effects and 2-SLS regressions to follow, we control for a player's age for this very reason as it affects baseline ability for a given player-season.

**Table 1. Summary statistics.**

| Variable | Mean | SD | Min | Median | Max |
|---|---|---|---|---|---|
| RPM | -0.82 | 2.57 | -8.44 | -1.17 | 9.79 |
| OtherPlayersRPM | -0.08 | 1.44 | -4.82 | -0.09 | 5.66 |
| Age | 27.03 | 4.25 | 19.08 | 26.50 | 42.00 |

N = 2,852 observations.

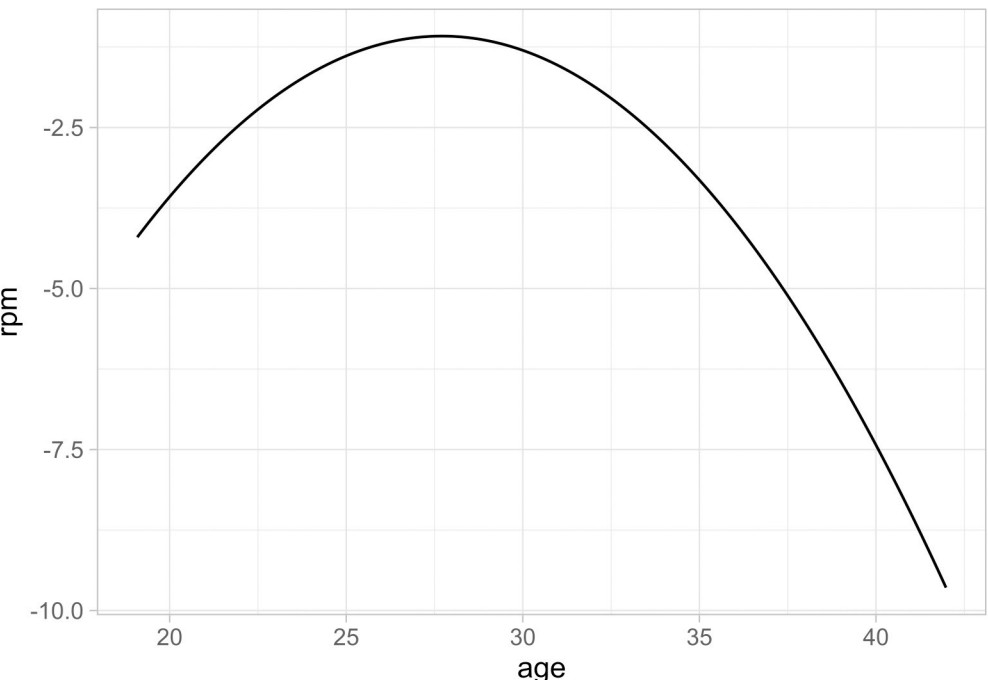

**Fig 6. Estimated real plus minus by age for the Typical NBA player.**

## III. Estimation approach

To answer our key question as to whether team member's RPM affects an individual player's RPM, we specify a baseline linear regression model with player RPM as a function of the player's individual characteristics such as age, age-squared, and minutes-weighted average RPM of lineup teammates.

Difficulties exist in estimating the relationship between player APM and lineup-teammate APM due to potential endogeneity. Just as a player's lineup-teammates may improve his APM value, so too might a player improve the APM values of his lineup-teammates. Moreover, players are often selected into lineups that potentially form a cluster of players with similar abilities. For example, starters disproportionately play alongside other starters, and backups disproportionately play alongside other backups. This latter issue also exacerbates endogeneity between player APM and lineup-teammate APM values. If we attempt to estimate the direct, uncontrolled relationship between these variables, we may actually pick up ability-clustering in lineup selections. We use the following techniques to address the endogeneity issue.

### a. Fixed effects estimation

One manner by which to treat endogeneity is through the use of fixed effects [11]. In the present setting, we take advantage of our six-year panel data structure to specify player fixed effects and treat endogeneity of these performance variables.

$$RPM_{i,t} = \beta_0 + \beta_1 \cdot player_i + \beta_2 \cdot season_t + \beta_3 \cdot OtherPlayersRPM_{i,t} + \beta_4 \cdot age_{i,t} + \beta_5 \cdot age_{i,t}^2$$
$$+ \varepsilon_{i,t} \tag{1}$$

where the response variable $RPM_{i,t}$ is the real plus-minus measure for player i in season t; $player_i$ and $season_t$ capture the player specific and time fixed effects, and OtherPlayersRPM

represents the RPM for the rest of the team members. We also control for the player's individual characteristics by age and its squared term. This information is available for 810 unique NBA players (i = 1 to 810) over 6 different seasons (t = 2014 to 2019) giving us 2852 total observations. This provides us the opportunity to control for individual specific as well as time specific fixed effects. There are 262 instances in which a player participated in more than one team in the same season. In that case, we average the player's RPM for that season. We perform a separate set of analyses using three-dimensional panel data (player i, team j, season t), but this dramatically reduces the total number of observations included in the final results. However, the final results do not change. To include the maximum possible observations in our analysis, we ignore the latter approach.

Player fixed effects control for unobserved baseline player ability such that the endogenous effect of the player's ability upon teammate performance is then isolated within the fixed effects model. With the specification of player fixed effects, one is implicitly regressing the performance of a player relative to his baseline (i.e., a performance residual for the player) against the APM level of lineup-teammates.

Given that we have specified player fixed effects herein, the fixed effects model specification in (1) represents an exogenous test as to whether RPM, a leading APM measure, is influenced by teammate productivity (i.e., through some complementarity effect that influences the overall size of the lineup "performance pie" and of the size of constituent slices).

## b. Two-stage least squares (2-SLS) estimation

Another approach we follow to subdue the problem of endogeneity is the use two-stage least squares (2-SLS) instrumental variable estimation using the lagged values of the minutes-weighted average APM of lineup teammates as an instrument. While it may be that lineup-teammate ability affects the player performance residual, there is a possibility that a player may influence the team's current period performance. To address such reverse causality, we use previous season performance of present season lineup teammates as an instrument for present quality of lineup teammates and use the estimated values in the regression as shown in model (2).

$$RPM_{i,t} = \delta_0 + \delta_1 \cdot \widehat{OtherPlayersRPM}_{i,t} + \delta_2 \cdot age_{i,t} + \delta_3 \cdot age_{i,t}^2 + \varepsilon_{i,t} \qquad (2)$$

where $\widehat{OtherPlayersRPM}$s the estimated values of other player RPM from the first stage regression. This technique follows the argument that present period teammate quality, as estimated by past performance of teammates, is exogenous of how a player performs in the current period and hence is a valid instrument for this estimation. This is a commonly used technique to subdue the problem of endogeneity [12]. The validity of the instrument is supported by the Cragg-Donald Wald F-statistic for the weak-identification test, as evidenced by the large enough F-statistics from the first stage regression. The NBA is a singular league in terms of competitive level that does not feature a great deal of in-season roster movement. Therefore, the best and perhaps only counterfactual measure of current player productivity in this case is lagged productivity of the same player. In sum, the result is that we can determine whether APM measures are truly teammate-independent within a model that estimates the relationship exogenously.

We regress model (1) and (2) for all NBA players and also in separate sub-samples by position-of-play for the Point Guard, Shooting Guard, Power Forward, Small Forward, and the Center positions. As such, we obtain six sets of regression results for each of the two estimation techniques explained above. We report the results in the subsequent section.

## IV. Empirical results

Estimation results are presented in Tables 2 and 3.

In the fixed effects regression results shown in Table 2, there is a significant, positive relationship between RPM and OtherPlayersRPM for the overall sample. For each unit of improvement in average lineup-teammate RPM, a player gains approximately 0.17 additional RPM points. As this result is conditional upon a given player's baseline productivity (via player fixed effects and age), we interpret this as a significant and fairly strong complementarity effect that is uncontrolled in the RPM measure. For each one standard deviation change in OtherPlayersRPM, a player gains an estimated 0.25 (*i.e.* 1.44 * 0.174) RPM units. Moreover, the estimated coefficient for OtherPlayersRPM is positive for each position-of-play in the position-of-play specific regressions, where the relationship is significant at standard p-values for the Small Forward and Center positions.

In the 2-SLS regressions results shown in Table 3, the sign of the coefficient for OtherPlayersRPM is positive and significant at the standard significance level for each regression. Whereas the fixed effects results for uncontrolled lineup-teammate complementarity are not significant for the PG, SG, and PF sub-samples, they are now statistically significant for all regressions. With respect to the size of the coefficients, they are each larger compared to those in the fixed effects model. In particular, for each unit of improvement in average lineup-teammate RPM, a player gains approximately 0.664 additional RPM points. Coefficients for other sub-samples can be interpreted accordingly.

## V. Discussion of the results

Based on the estimations above, for each unit average gain in the teammate's RPM, a player's RPM is overestimated by a range of 0.17 to 0.66 points according to point estimates. We find that RPM is not context-independent. Rather, positive spillover effects from more capable teammates improve a given player's RPM value, ceteris paribus. This result has implications upon player trade value. Namely, a player's trade value is estimated to be a function of where he is traded. Indeed, basketball is a game of complementarity and no measure—not even a measure such as RPM—has demonstrated the ability to isolate these complementarity effects toward an understanding of player value in the abstract. While this is perhaps the holy grail of

**Table 2. The fixed effects estimation results.** Response Variable: Individual Player RPM.

| | All | PG | SG | SF | PF | C |
|---|---|---|---|---|---|---|
| Other RPM | 0.17*** | 0.12 | 0.12 | 0.27*** | 0.11 | 0.27*** |
| | (0.036) | (0.087) | (0.079) | (0.10) | (0.083) | (0.072) |
| Age | 2.55*** | 1.92*** | 2.54*** | 2.58*** | 3.09*** | 3.09*** |
| | (0.23) | (0.51) | (0.50) | (0.64) | (0.63) | (0.65) |
| Age Squared | -0.046*** | -0.036*** | -0.047*** | -0.046*** | -0.058*** | -0.057*** |
| | (0.0034) | (0.0081) | (0.0076) | (0.0091) | (0.0084) | (0.0079) |
| Constant | -34.8*** | -25.5*** | -34.6*** | -35.8*** | -40.6*** | -41.3*** |
| | (4.27) | (8.61) | (8.59) | (11.9) | (12.1) | (13.2) |
| N | 2590 | 516 | 580 | 464 | 533 | 497 |
| $R^2$ | 0.12 | 0.10 | 0.13 | 0.14 | 0.16 | 0.20 |

PG = Point Guard; SG = Shooting Guard, PF = Power Forward, SF = Small Forward, C = Center positions. Standard errors in parentheses;

* $p < 0.10$,

** $p < 0.05$,

*** $p < 0.01$.

**Table 3. Instrumental variable regression.** Response Variable: Individual Player RPM.

|  | **All** | **PG** | **SG** | **SF** | **PF** | **C** |
|---|---|---|---|---|---|---|
| Other RPM | 0.664*** | 1.185*** | 0.716* | 0.797*** | 0.715*** | 0.548* |
|  | (0.138) | (0.250) | (0.423) | (0.242) | (0.261) | (0.291) |
| Age | 0.696*** | 1.371*** | 0.703* | 1.058*** | 0.656* | 0.159 |
|  | (0.173) | (0.474) | (0.365) | (0.410) | (0.356) | (0.366) |
| Age Squared | -0.0122*** | -0.0245*** | -0.0122* | -0.0181*** | -0.0114* | -0.00298 |
|  | (0.00299) | (0.00824) | (0.00628) | (0.00698) | (0.00608) | (0.00633) |
| Constant | -10.35*** | -19.30*** | -11.11** | -15.79*** | -9.510* | -1.822 |
|  | (2.468) | (6.739) | (5.201) | (5.893) | (5.143) | (5.217) |
| N | 1729 | 339 | 372 | 304 | 359 | 355 |
| $R^2$ | 0.11 | 0.17 | 0.10 | 0.11 | 0.16 | 0.11 |
| 1st Stage F-Stat | 186.58 | 92.00 | 42.39 | 55.00 | 31.53 | 23.27 |

PG = Point Guard; SG = Shooting Guard, PF = Power Forward, SF = Small Forward, C = Center positions. F-statistic represent the Cragg-Donald Wald F-statistic for the weak-instrument test. The statistics are large enough to reject the null hypothesis of weakly identified equation. Standard errors in parentheses;

* $p < 0.10$,

** $p < 0.05$,

*** $p < 0.01$.

basketball data science, it is not assured that such a measure is possible. As perhaps similar to quantum mechanical information, it may be impossible to understand the contribution of the basketball player and that of the player's environment in some absolute sense. Indeed, basketball leagues are not natural experiments in which players are randomly paired and resampled. Rather, players are organized into often stable team environments and resampling occurs infrequently such that players have often aged by the time they receive a new set of teammates with whom to play. In such an environment, counterfactuals concerning player value will go largely unobserved.

## VI. Conclusions

The results provide strong evidence that regularized adjusted plus minus player productivity measures are not, in fact, "teammate-independent." Rather, we find evidence that lineup-teammate productivity positively influences a given player's real plus minus value. As this result is conditional upon a given player's baseline productivity via player fixed effects and age, we interpret this as a significant and fairly strong complementarity effect that is uncontrolled in adjusted plus minus measures such as real plus minus. While real plus minus may control for in-sample teammate effects well, it appears that the measure does not control for out-of-sample lineup-teammate quality effects. We find this within a model that accounts for teammate quality changes from season to season. We note that basketball leagues are not natural experiments in which players are randomly paired and resampled. Rather, players are organized into often stable team environments and resampling occurs infrequently such that players have often aged by the time they receive a new set of teammates with whom to play. In such an environment, counterfactuals concerning player value will go largely unobserved. From this estimation, further (out-of-sample) adjustments to the APM estimation methodology can be explored in future work.

## Author Contributions

**Conceptualization:** Shankar Ghimire, Justin A. Ehrlich, Shane D. Sanders.

**Data curation:** Shankar Ghimire, Justin A. Ehrlich, Shane D. Sanders.

**Formal analysis:** Shankar Ghimire, Justin A. Ehrlich, Shane D. Sanders.

**Methodology:** Shankar Ghimire, Justin A. Ehrlich, Shane D. Sanders.

**Resources:** Shane D. Sanders.

**Software:** Shankar Ghimire, Justin A. Ehrlich, Shane D. Sanders.

**Validation:** Shankar Ghimire.

**Visualization:** Shankar Ghimire, Justin A. Ehrlich, Shane D. Sanders.

**Writing – original draft:** Shankar Ghimire, Justin A. Ehrlich, Shane D. Sanders.

**Writing – review & editing:** Shankar Ghimire, Justin A. Ehrlich, Shane D. Sanders.

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
