## [Decision Letter · Decision Letter 0]

25 Jun 2020

PONE-D-20-05517

Measuring Individual Worker Output in a Complementary Team Setting: Does Regularized Adjusted Plus Minus Isolate Individual NBA Player Contributions?

PLOS ONE

Dear Dr. Ghimire,

Thank you for submitting your manuscript to PLOS ONE. After careful consideration, we feel that it has merit but does not fully meet PLOS ONE’s publication criteria as it currently stands. Therefore, we invite you to submit a revised version of the manuscript that addresses the points raised during the review process.

We look forward to receiving your revised manuscript.

Kind regards,

Corrado Andini

Academic Editor

PLOS ONE

Journal Requirements:

Additional Editor Comments (if provided):

The identification of a causal effect in the paper requires a valid instrument. The use of lagged values as instruments is a bit out of fashion for several reasons proposed in the panel-data literature, which the authors should discuss. The authors should make an effort to provide additional arguments in support of their identification approach, also keeping into account the comments of the referees.

Reviewers' comments:

Reviewer's Responses to Questions

**Comments to the Author**

1. Is the manuscript technically sound, and do the data support the conclusions?

Reviewer #1: Yes

Reviewer #2: Partly

2. Has the statistical analysis been performed appropriately and rigorously? 

Reviewer #1: Yes

Reviewer #2: Yes

3. Have the authors made all data underlying the findings in their manuscript fully available?

Reviewer #1: Yes

Reviewer #2: Yes

4. Is the manuscript presented in an intelligible fashion and written in standard English?

Reviewer #1: Yes

Reviewer #2: Yes

5. Review Comments to the Author

Reviewer #1: The paper is well-written and discusses the endogeneity issue appropriately, but I think some changes have the potential to improve the manuscript. Here are a few comments.

The data reported in Figure 3 show that there have been less rookies in 2016. Rookies increased by 7.5 percentage points the year after, representing a 50% raise. Is there a reason for this raise?

I would be cautious in deriving a general formula for adjusted APM measure, i.e., equation 2. While the empirical analysis is performed appropriately, the estimates are based on data restricted to 4 time periods.

The validity of the instrument could be supported by the results of the Sargan statistic and the Ftest.

Figures should be self-explanatory. I suggest adding notes to describe what is plotted. Also, the legend in Figure 5 is not clear as the acronyms C, PF, PG, SF, and SG were not defined before.

Abbreviations should be defined the first time they appear.

On a minor note, it may be interesting to see what happens if the authors exclude the observations closer to the maximum age, for which the productivity is much lower.

Reviewer #2: This paper contests the claim that "Adjusted plus minus (APM)" measures are independent from teammate productivity. In order to show that APM player productivity measures are indeed dependent from the one of other team players, the authors perform an empirical analysis using open data from NBA player-season, Real Plus Minus for all recorded player-seasons from 2013-19, and player lineup data.

- The research question is clearly explained and contextualized.

- As it regards the estimation approach, I suggest showing the equation for both model specifications (a and b).

- In the “Two-stage Least Squares (2-SLS) Estimation” the goodness of the proposed instrument is not discussed. Conversely, it is important to show that the suggested instrument is not a weak one. Hence, I suggest including the F-statistic at least in the results’ table.

- In the fifth section I would suggest a more general discussion of the results than the derivation of a new measure of the individual player performance.

- I do not agree on the derivation of the adjusted APM measure (equation 2). The coefficient of the variable “OtherPlayersRPMi,t” is roughly the mean of the previous "OtherPlayersRPMi,t"’s estimated coefficients in the two separated models (with different underlying assumptions). I have serious doubts on its validity.

Minor comments

Notes about the tables are missing. The abbreviations used in the tables should be explained below the tables.

6. PLOS authors have the option to publish the peer review history of their article (what does this mean?). If published, this will include your full peer review and any attached files.

Reviewer #1: No

Reviewer #2: No

---

## [Author Response · Author response to Decision Letter 0]

17 Jul 2020

Editor's Comment: 

In the paragraphs below, we address the concerns raised by the reviewers on a comment-by-comment basis. The changes are highlighted in red within the manuscript. With respect to your own feedback on the use of instrument, the validity of the instrument is supported by the Cragg-Donald Wald F-statistic for the weak-identification test, as evidenced by the large enough F-statistics from the first stage regression. We agree with your comment that it would be ideal to use and external instrument, but we resort to the lagged-productivity technique in the absence of external instrument while also meeting our objective of checking the robustness of our results from the fixed effects estimation. The NBA is a singular league in terms of competitive level that does not feature a great deal of in-season roster movement. Therefore, the best and perhaps only counterfactual measure of current player productivity in this case is lagged productivity of the same player. This discussion has been added in the estimation section (page 10). 

Reviewer #1: The paper is well-written and discusses the endogeneity issue appropriately, but I think some changes have the potential to improve the manuscript. Here are a few comments.

Response: Thank you for your comments. We appreciate your feedback. Below, we address your concerns on a comment-by-comment basis. The changes are highlighted in red within the manuscript.

1. The data reported in Figure 3 show that there have been less rookies in 2016. Rookies increased by 7.5 percentage points the year after, representing a 50% raise. Is there a reason for this raise?

Response: In 2016 there was roster expansion from 15 to 17 per team given the advent of 2-way contracts. This allowed additional rookies to gain limited NBA exposure during 2016. Specifically, several second round draft picks and undrafted free agent rookies who might have otherwise spent the season in the G-League were signed to two-way contracts for the first time in 2016. 

2. I would be cautious in deriving a general formula for adjusted APM measure, i.e., equation 2. While the empirical analysis is performed appropriately, the estimates are based on data restricted to 4 time periods.

Response: Upon review of your valuable comment, we agree and have striken this from the analysis. 

3. The validity of the instrument could be supported by the results of the Sargan statistic and the F-test.

Response: 

Table 3 has been updated to reflect the test statistics. Our estimation has only one endogenous regressor and one external instrumental variable. Accordingly, we report the F-statistic. The validity of the instrument is supported by the Cragg-Donald Wald F-statistic for the weak-identification test, as evidenced by the large enough F-statistics from the first stage regression. Additional explanation has been added in the estimation section (page 10). 

4. Figures should be self-explanatory. I suggest adding notes to describe what is plotted. Also, the legend in Figure 5 is not clear as the acronyms C, PF, PG, SF, and SG were not defined before.

Abbreviations should be defined the first time they appear.

Response: The Figure 5 caption has been updated to reflect the acronyms used in the paper. 

5. On a minor note, it may be interesting to see what happens if the authors exclude the observations closer to the maximum age, for which the productivity is much lower.

Response:

We ran the regressions excluding those players that are above the age of 40. There are only six such players and they do not significantly alter the results. 

Table 3: Instrumental Variable Regression: Lag of other player's RPM as an instrument Age 

 All PG SG SF PF C

Other's RPM 0.669*** 1.187*** 0.695** 0.670** 0.743** 0.531*

 (0.134) (0.257) (0.334) (0.273) (0.316) (0.289)

Age 0.739*** 1.372*** 0.726** 1.245*** 0.731* 0.149

 (0.179) (0.477) (0.359) (0.448) (0.423) (0.363)

Age Squared -0.0130*** -0.0246*** -0.0126** -0.0217*** -0.0131* -0.00278

 (0.00309) (0.00828) (0.00622) (0.00769) (0.00734) (0.00628)

Constant -10.93*** -19.29*** -11.44** -18.31*** -10.30* -1.685

 (2.537) (6.774) (5.102) (6.386) (6.025) (5.174)

N 1723 339 371 300 356 357

Note. PG = Point Guard; SG = Shooting Guard, PF = Power Forward, SF = Small Forward, C = Center positions. Standard errors in parentheses; * p < 0.10, ** p < 0.05, *** p < 0.01. 

 

Reviewer #2: This paper contests the claim that "Adjusted plus minus (APM)" measures are independent from teammate productivity. In order to show that APM player productivity measures are indeed dependent from the one of other team players, the authors perform an empirical analysis using open data from NBA player-season, Real Plus Minus for all recorded player-seasons from 2013-19, and player lineup data. The research question is clearly explained and contextualized.

Thank you for your comments. We appreciate your feedback. Below we address your concerns comments by comments. The changes are highlighted in red within the manuscript.

1. As it regards the estimation approach, I suggest showing the equation for both model specifications (a and b).

Response:

Section III has been re-organized to include the regression equations separately for the fixed effects model and the 2SLS model. 

2. In the “Two-stage Least Squares (2-SLS) Estimation” the goodness of the proposed instrument is not discussed. Conversely, it is important to show that the suggested instrument is not a weak one. Hence, I suggest including the F-statistic at least in the results’ table.

Response: 

Table 3 has been updated to reflect the test statistics. Our estimation has only one endogenous regressor and one external instrumental variable. Accordingly, we report the F-statistic. The validity of the instrument is supported by the Cragg-Donald Wald F-statistic for the weak-identification test, as evidenced by the large enough F-statistics from the first stage regression. Additional explanation has been added in the estimation section (page 10).

3. In the fifth section I would suggest a more general discussion of the results than the derivation of a new measure of the individual player performance.

a. I do not agree on the derivation of the adjusted APM measure (equation 2). The coefficient of the variable “OtherPlayersRPMi,t” is roughly the mean of the previous "OtherPlayersRPMi,t"’s estimated coefficients in the two separated models (with different underlying assumptions). I have serious doubts on its validity.

Response: Upon reflection, we agree and have stricken equation (2). Moreover, we now provide an extended general discussion of the results and their importance/interpretation as follows: 

Added passage (page 13-14): 

“We find that RPM is not context-independent. Rather, positive spillover effects from more capable teammates improve a given player’s RPM value, ceteris paribus. This result has implications upon player trade value. Namely, a player’s trade value is estimated to be a function of where he is traded. Indeed, basketball is a game of complementarity and no measure—not even a measure such as RPM—has demonstrated the ability to isolate these complementarity effects toward an understanding of player value in the abstract. While this is perhaps the holy grail of basketball data science, it is not assured that such a measure is possible. As perhaps similar to quantum mechanical information, it may be impossible to understand the contribution of the basketball player and that of the player’s environment in some absolute sense. Indeed, basketball leagues are not natural experiments in which players are randomly paired and resampled. Rather, players are organized into often stable team environments and resampling occurs infrequently such that players have often aged by the time they receive a new set of teammates with whom to play. In such an environment, counterfactuals concerning player value will go largely unoberserved.”

4. Minor comments: Notes about the tables are missing. The abbreviations used in the tables should be explained below the tables.

Response: Notes added at the bottom of Table 2 and Table 3 to reflect the full forms of the abbreviations.

---

## [Editor Report · Decision Letter 1]

6 Aug 2020

Measuring Individual Worker Output in a Complementary Team Setting: Does Regularized Adjusted Plus Minus Isolate Individual NBA Player Contributions?

PONE-D-20-05517R1

Dear Dr. Ghimire,

We’re pleased to inform you that your manuscript has been judged scientifically suitable for publication and will be formally accepted for publication once it meets all outstanding technical requirements.

Kind regards,

Corrado Andini

Academic Editor

PLOS ONE
---

## [Editor Report · Acceptance letter]

12 Aug 2020

PONE-D-20-05517R1 

Measuring Individual Worker Output in a Complementary Team Setting: Does Regularized Adjusted Plus Minus Isolate Individual NBA Player Contributions? 

Dear Dr. Ghimire:

I'm pleased to inform you that your manuscript has been deemed suitable for publication in PLOS ONE. Congratulations! Your manuscript is now with our production department. 

Kind regards, 

on behalf of

Professor Corrado Andini 

Academic Editor

PLOS ONE